**communications** engineering

# Object detection as an aid for locating the prostate in surface-based abdominal ultrasound images
Rory D. Bennett[1], Tristan Barrett[2], Vincent J. Gnanapragasam[3] & Zion Tsz Ho Tse [1] ✉

Automatic object detection is increasingly used in the medical field to enhance clinical workflows before, during, and after diagnosis of various conditions. One example is prostate detection and prostate volume estimation, which can aid in triaging patients for prostate cancer through risk-stratification using prostate-specific antigen density. In this paper, a baseline prostate detection framework is presented, highlighting that current state-of-the-art object detection models can detect the prostate in difficult to interpret surface-based ultrasound images. A 5-fold cross-validation study returned intersection-over-union, precision, recall, F1, and average-precision values above 0.7 with real-time capabilities possible. Additionally, a simple size calculation based on the detection results showed high correlation with ground truth measurements, with Pearson Correlation Coefficients ranging from 0.55 to 0.84 for prostate volume estimates. These findings will contribute to the development of a real-time prostate detection and size estimation platform for prostate cancer risk-stratification to reduce unnecessary biopsy rates in healthcare systems.

Prostate cancer (PCa) affects 1 in 8 men in the United Kingdom annually[1], with a mortality rate second only to lung cancer (in men)[2]. The 5-year survivability rate drop between those diagnosed with stage 3 and stage 4 cancer highlights the importance of earlier detection (95% down to 50%, respectively[3]). The current diagnostic work-up of PCa involves prostate-specific antigen (PSA) testing and magnetic resonance imaging (MRI) scanning, however, screening for PCa has proven unsuccessful due to the non-specific nature of PSA[4,5]. MRI combined with target biopsies has been proposed to overcome this non-specific nature while limiting costs and side-effects[6]. However, MRI is still an expensive and inefficient way to diagnose PCa. Therefore, a filter/triage test is highly desirable prior to performing an MRI. Indeed, this is one of the main focuses of major PCa early detection efforts as exemplified by the European Union PRAISE-U initiative[7]. An important proposed tool is the use of PSA-density (PSAD) which combines the results of a PSA blood test with prostatic volume (PV), normalising PSA with PV. PSAD accounts for larger prostates naturally producing more PSA, with previous studies concluding that PSAD is a much more reliable PCa indicator than PSA alone[8–10]. As such, PSAD is emerging as a helpful bio-marker for early PCa detection, augmenting biopsy decision making in conjunction with MRI and in cancer management, e.g. active surveillance programmes[11,12].

The clinical approach for acquiring PV typically relies on either MRI or transrectal ultrasound (TRUS) imaging. However, these modalities are subject to higher operating costs (MRI) and increased patient discomfort (TRUS). A newer and more patient acceptable modality is that of surface-based ultrasound (SUS) which, when used to infer PV for calculation of PSAD, is generally considered accurate enough in comparison to more contemporary methods – without being subject to the same disadvantages[13]. Deriving PV from SUS images can be split into two distinct problems. The first is locating the prostate and the second is determining the PV. Both steps currently require a skilled clinician, limiting widespread use of SUS-derived PSAD for PCa risk stratification/ screening/triaging. In this study, we show that the first problem (prostate detection in SUS images) can be largely addressed using current state-of-the-art (SOTA) deep learning object detection algorithms with real-time application possible. We also show that the object detection results can be used to estimate PV with high correlation to ground truth values with a simple calculation based on inferred bounding box dimensions.

Three SOTA deep learning object detection algorithms were compared in this study: You Only Look Once (YOLO) version 8 (most recent version at the commencement of this study)[14], RetinaNet[15], and FasterRCNN[16]. These three algorithms were chosen as they have

[1]School of Engineering and Materials Science, Queen Mary University of London, Mile End Road, London, E1 4NS, UK. [2]Department of Radiology, University of Cambridge School of Clinical Medicine, Cambridge, CB2 0QQ, UK. [3]Department of Surgery, University of Cambridge School of Clinical Medicine, CB2 0QQ Cambridge, UK. ✉e-mail: z.tse@qmul.ac.uk

**Fig. 1 | Potential PCa (prostate cancer) screening/ risk-stratification workflow.** A patient presents for screening. Blood is drawn (for a PSA (prostate-specific antigen) blood test) and a SUS (surface-based ultrasound) scan of the prostate is conducted using a point-of-care ultrasound system. The scan is used to calculate PV (prostate volume) which is combined with the PSA result to give PSAD (prostate-specific antigen density); *ng* (nanograms); *ml* (millilitres). Further testing options (MRI (magnetic-resonance imaging) needle biopsy) are discussed with the patient if necessary.

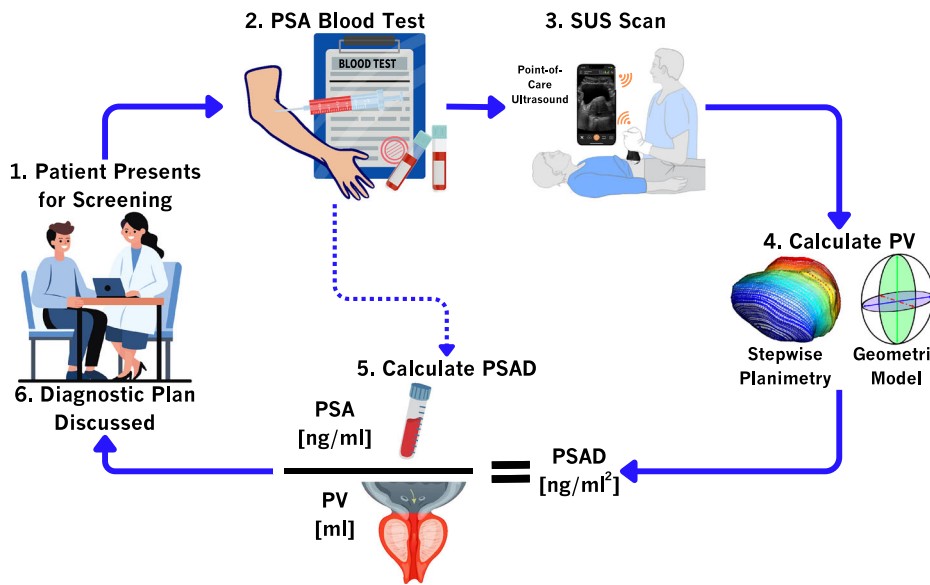

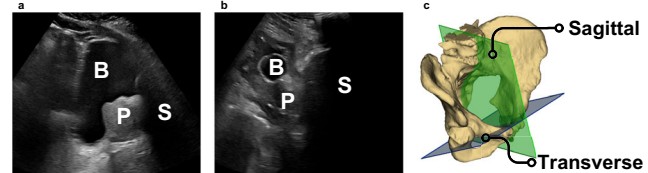

relatively fast inference speeds, which is important for real-time detection. Real-time detection allows for immediate feedback to the ultrasound probe operator indicating whether the prostate is in view. Such feedback will be particularly important in SUS scans of the prostate, as the images can be difficult to interpret. All three algorithms have had their utility validated in various applications[17–20], with object detection being a well-researched field across myriad domains. However, to the best of the authors knowledge, deep learning guided prostate detection using SUS images has not been investigated before. While this speaks to the novelty of the work presented, it does make comparisons to previous studies difficult. Therefore, the object detection results are presented such that they can be used as a baseline for comparisons in future studies.

Figure 1 serves to highlight a proposed triage/screening/risk-stratification workflow, from a patient presenting at their primary care institution to them undergoing a SUS scan of the prostate for PSAD calculations. Step 3 of this workflow is the focus of this study, with a brief consideration of Step 4. Before calculating PV, an adequate SUS scan of the prostate is required. Presently, this requires a highly skilled clinician to conduct the scan as the prostate can be difficult to locate. With the help of machine learning and AI a highly skilled clinician may no longer be required to scan the prostate. By showing that current real-time SOTA object detection algorithms can locate the prostate in SUS images of the prostate to a relatively high degree of accuracy, a system can be developed that gives ultrasound probe operators immediate feedback. This can ensure adequate SUS scans of the prostate for PSAD calculations, even from clinicians with minimal training.

One previous study has attempted to estimate prostate size from SUS images of the prostate[21]. The study described an end-to-end system for estimating PV from SUS images by inferring key anatomical landmarks. Specifically, four points on transverse transabdominal ultrasound (tAUS) images and two points on sagittal transabdominal ultrasound (sAUS) images, totalling six key points to define three orthogonal dimensions – see Fig. 2c for reference scan planes. A quadruplet deep convolutional network (QDCNN) was developed to estimate these points which were then converted to the three major dimensions used in Equation (5) to calculate PV. While it could be argued that successful identification of these landmarks implies the presence of the prostate in the image, this is conceptually distinct from object detection methods, which typically aim to localise regions of interest via bounding boxes and assigned class probabilities. Their QDCNN system performed exceptionally well at calculating the size of the prostate (in line with expert estimates), however, it was not real-time and could not be used to help guide an ultrasound probe operator towards the prostate during the scanning process.

**Fig. 2 | Sample AUS (transabdominal ultrasound) images and reference scan planes.** Sample images from the prospective dataset (**a**) and retrospective dataset (**b**), both taken in the sagittal plane with "B" indicating bladder (or bladder location), "P" the prostate, and "S" pubic bone shadowing. The prospective image shows standard anatomical features (bladder, prostate, shadowing) while the retrospective image shows a patient who has had a catheter inserted into their bladder. Reference scan planes are shown in **c**.

In this study it has been shown that current SOTA object detection algorithms are capable of real-time (or near real-time) prostate detection with high accuracy. The detection results (boxes) can also be used in a rudimentary size calculation to estimate PVs that show moderate to high correlation to ground truth values.

## Results

### Cross-validation results

A summary of the results of the 5-fold cross-validation study are given in Fig. 3 for each of the trained models across all metrics considered. Associated numerical values can be found in Supplementary Table 3. A slight drop between the IoU* metric and the IoU metric can be seen across all models. Otherwise, there was only minor variation between models. All models scored above 0.7 for the IoU* metric, and above 0.6 for all other metrics except $AP_m75$ and $AP_m50$-95. The $AP_m75$ and $AP_m50$-95 metrics are a more stringent $AP_m50$, which is highlighted by the drop in performance for the two metrics across all models. The RetinaNet[P] model performed marginally better across most metrics, followed by both YOLO models, then the FasterRCNN[P] model, RetinaNet[PB] model, and finally the FasterRCNN[PB] model. For all models, the prostate only detection versions performed better that the prostate and bladder detection versions.

Figure 4 shows sample results from each model on the same sAUS image (left) and tAUS image (right) with ground truth bounding boxes also given. While the highest confidence boxes' locations were similar across all models, RetinaNet and FasterRCNN models generally showed higher confidence values than the YOLO models. The average confidence of each

**Fig. 3 | 5-fold cross-validation metric results.** All metrics were calculated for all detected prostate bounding boxes except IoU*. The dashed lines correspond to models trained to detect both bladder and prostate (superscript PB), with solid lines corresponding to models that were trained to detect prostate only (superscript P). IoU* (Intersection-over-union of highest confidence prostate bounding box); IoU (Intersection-over-union of all detected prostate boxes); AP (Average precision at varying thresholds).

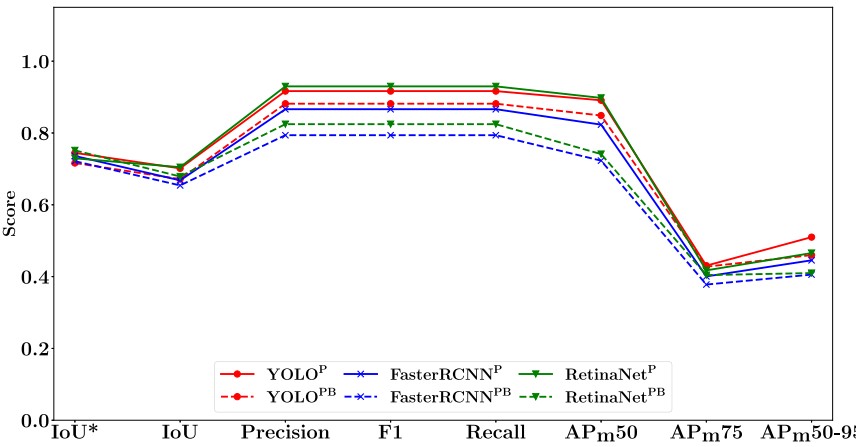

model across all patients for the highest confidence bounding box is given in Supplementary Table 3.

Figure 5 shows the PCCs and RMSEs when comparing the ground truth dimensions to the model inferred dimensions. Corresponding numerical values (dimension PCCs and RMSEs) can be found in the Supplementary Table 4. For the SI dimension, lowest correlation was noted for the YOLO$^{PB}$ model, with half of the models showing low correlation. For the RL dimension, only the FasterRCNN$^{PB}$ model had a low correlation with the YOLO$^P$ model showing the highest correlation. For the AP dimension, all models had a high correlation, with YOLO$^P$ having the highest correlation. For the volume measurement, only YOLO$^{PB}$ had a low correlation, with RetinaNet$^{PB}$ showing the highest correlation of 0.84. The means of the SI, RL, AP, and volume correlations across all models were calculated as 0.5, 0.62, 0.7, and 0.74, respectively. A similar pattern was noted for all models when considering the RMSE values. The RL dimension showed the lowest RMSE values, followed by the AP dimension, then the SI dimension, and finally the volume. For the SI and RL dimensions, the RentinaNet$^{PB}$ model showed the lowest RMSE; for the AP dimension the YOLO$^P$ model showed the lowest RMSE; and for the volume measurement the RetinaNet$^{PB}$ model showed the lowest RMSE.

The BA plots in Supplementary Figs. 1 and 2 highlight the tendency of all the models to overestimate the volume of the prostate, in both $cm^3$ and error percentage, as indicated by the relatively large positive bias. The range of the LoAs (across all models) varied from $140cm^3$ to $245cm^3$, or 239% to 391%, with very few measurements falling within the more acceptable error range of $\pm 15cm^3$ from the mean difference (this acceptable error range is fairly arbitrary, and was used in a previous study[22]). To address the large LoAs, a BA analysis was conducted where the SI dimension was halved for all models, the plots of which are given in Supplementary Fig. 3. This adjustment reduced the bias across all models. The range of the LoAs was also reduced to $69cm^3$ to $130cm^3$, or 119% to 195%, with more measurements falling within the acceptable error range.

The average inference times for a single image for each model are given in Table 1. YOLO had the lowest inference speed (highest frames-per-second), followed by RetinaNet, with FasterRCNN showing the slowest inference speed.

Figure 6 serves to highlight the importance of the minimum confidence threshold of 0.3 that was applied to all models. The figure shows what was returned by the RetinaNet$^{PB}$ model for varying thresholds, from no threshold (0) up to the chosen threshold of 0.3. If the threshold was too low inferred bounding boxes were scattered across the image, almost randomly, for both classes (prostate and bladder). Increasing the threshold resulted in a substantial drop in inferred bounding box scattering, with a threshold of 0.3 removing most unwanted bounding boxes from the models' output.

**Retrospective results**

The results obtained when testing the models on the retrospective dataset is split into two sections. The first summarises the results of the non-special

cases (22 patients in total). This is followed by a brief look at how the models performed on the special cases only.

Figure 7 summarises the metric scores for both sets of models (prospective and IPV) with corresponding numerical values given in Supplementary Tables 5 and 6. For the prospective models, both RetinaNet models scored highest in most metrics, followed by the YOLO$^{PB}$ and FasterRCNN$^{PB}$ models, then the YOLO$^P$ and FasterRCNN$^P$ models. All models showed a drop in performance in the AP$_m$75 and AP$_m$50-95 metrics. For the IPV metrics, the YOLO models scored the highest in most metrics, followed by the FasterRCNN models and RetinaNet models.

Figure 8 highlights the improvement between the prospective YOLO$^{PB}$ model and the IPV YOLO$^{PB}$ model, with the left column showing cases where the YOLO$^{PB}$ model failed to detect the prostate or detected a box with a very low IoU*, while the right column shows all-round improvements in detection and IoU*.

Figure 9 summarises PCC and RMSE results. Corresponding numerical results can be found in Supplementary Tables 7 and 8. For the prospective models' PCCs (top left), only the FaterRCNN$^{PB}$ and YOLO$^P$ models showed high correlation for the SI dimension; half of the models showed a high correlation for the RL dimension; all models showed a low correlation for the AP dimension; and only the FasterRCNN$^P$ model showed a low correlation for the volume measurement. The means of the SI, RL, AP, and volume correlations across all models were calculated as 0.51, 0.59, 0.3, and 0.69, respectively. For the IPV models' PCCs (top right), the YOLO$^{PB}$ and FasterRCNN$^{PB}$ models showed high correlation for the SI dimension; only the RetinaNet models showed high correlation for the RL dimension; half of the models showed high correlation for the AP dimension; with all models showing a high correlation for the volume measurement. The means of the SI, RL, AP, and volume correlations across all models were calculated as 0.57, 0.52, 0.52, and 0.84, respectively. For both the prospective and IPV models' RMSEs (bottom left and right), a similar pattern was noted across the models. The RL dimension showed the lowest RMSE, followed by the AP dimension, SI dimension, and finally the volume.

Sample results, metrics, and dimension correlation and RMSE plots for the special cases are given in Supplementary Figs. 4, 5, and 6, with corresponding numerical values given in Supplementary Tables 9 and 10. A noticeable drop across all metrics was noted for prospective and IPV models (in comparison to non-special cases), with most model measurements showing low correlation with expert markings. The prospective RetinaNet models and the IPV FasterRCNN$^{PB}$ and YOLO$^{PB}$ models showed the best performance for most metrics, while the prospective and IPV FasterRCNN$^{PB}$ model generally had the highest dimension correlations.

**Discussion**

**Cross-validation study**

The results of the cross-validation study showed that all three models performed exceptionally well in detecting the prostate. It was noted that if the confidence threshold was not applied, RetinaNet showed a substantial drop

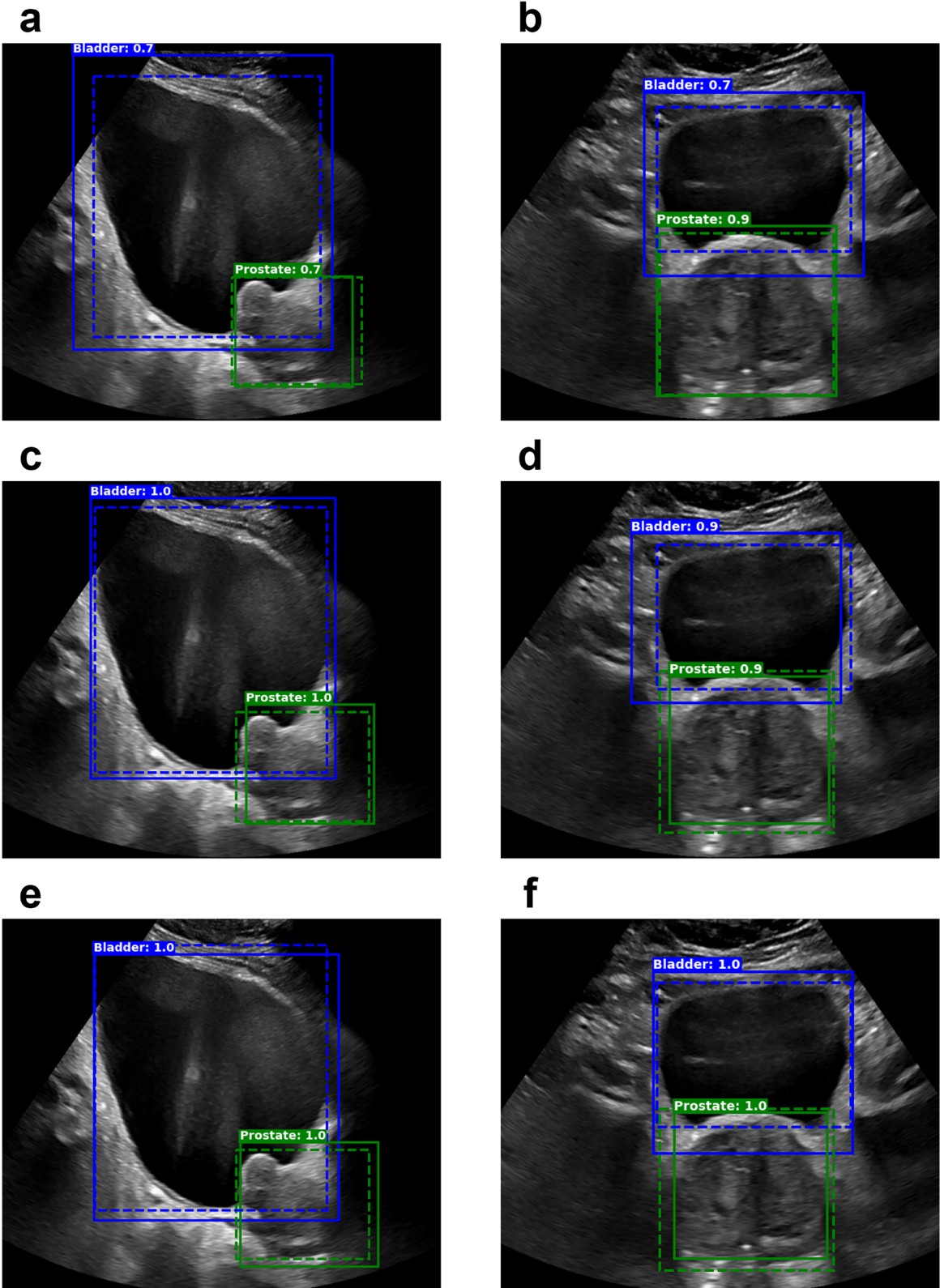

**Fig. 4 | Sample cross-validation object detection results for the SOTA (state-of-the-art) detection models.** Results from the YOLO (**a**, **b**), RetinaNet (**c**, **d**), and FasterRCNN (**e**, **f**) models for sAUS (sagittal transabdominal ultrasound) (left column) and tAUS (transverse transabdominal ultrasound) (right column) images. Model inferred boxes are solid lines, while ground truth boxes are dashed. Only the highest confidence boxes are shown, with green for prostate and blue for bladder. White text indicates box confidence.

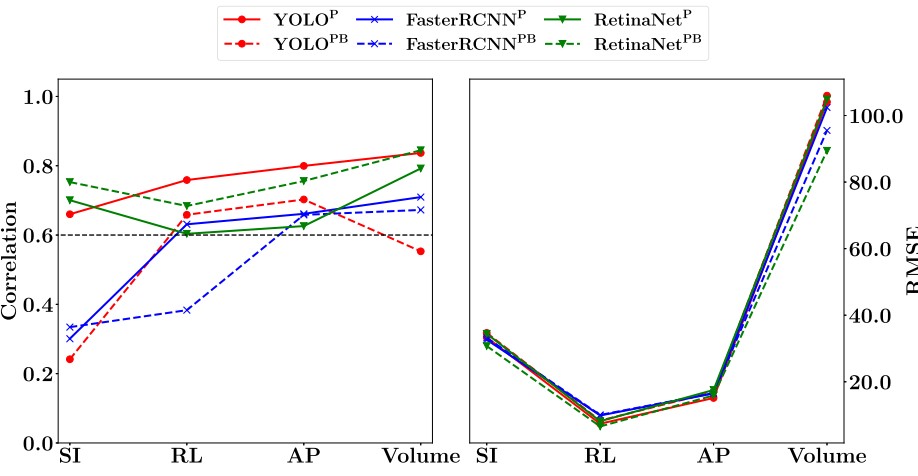

**Fig. 5 | Cross-validation dimension correlations and error results.** PCCs (Pearson correlation coefficients) (left) and RMSEs (root-mean-squared-errors) (right) comparing ground truth dimensions with bounding box inferred dimensions for the YOLO (red), RetinaNet (green), and FasterRCNN (blue) models. The dashed lines correspond to models trained to detect both bladder and prostate (superscript PB), with solid lines corresponding to models that were trained to detect prostate only (superscript P). Only the highest confidence bounding box was used for the inferred dimension calculations. The black line indicates a PCC of 0.6. SI (superior-inferior), RL (right-left), and AP (anterior-posterior) RMSE values are in mm while the Volume RMSE value is in cm³.

**Table 1 | Inference speeds in milliseconds (ms) per image and relative frames-per-second (fps) for each model during testing**

|  | YOLO | RetinaNet | FasterRCNN |
|---|---|---|---|
| **Milliseconds Per Image** | 43 | 57 | 137 |
| **Frames Per Second** | 23.3 | 17.5 | 7.3 |

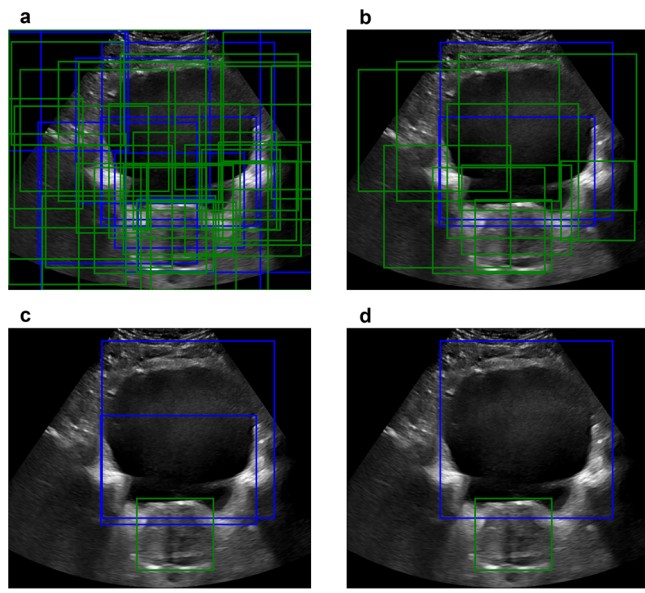

**Fig. 6 | Cross-validation RetinaNet^PB model results with varying confidence thresholds. a–d** Correspond to threshold values of 0, 0.1, 0.2, and 0.3, respectively. Blue and green boxes indicate model detected bladder and prostate boxes, respectively.

in performance. This was due to RetinaNet still returning multiple low confidence boxes that were scattered over the input image (Fig. 6). However, in this specific use-case – where only a single object of a given class is present in the image – this spread of detected boxes might not necessarily present itself as a problem as only the highest confidence box would be considered in further calculations. Therefore, considering IoU* as the primary performance metric means all models performed about the same, with RetinaNet^PB performing only marginally better than the other models. An interesting result from the metric scores was that the prostate only detection models showed higher metric scores than their prostate and bladder detection counterparts. A possible reason for this is that increasing the number of detectable classes for a model increases complexity as the model needs to learn more features/class representations to separate each class. The limited dataset size of this study prevents a full analysis of this phenomenon, and future work could test if the same results persisted with more data.

While the individual dimensions' correlation values varied between very low (SI – YOLO^PB, FasterRCNN^P, FasterRCNN^PB) and quite high (RL and AP – all models except FasterRCNN^PB), the volume measurement correlation tended to be high (>0.6). The dimension with the lowest average correlation across models was the SI dimension. This was expected, as sAUS images are typically more difficult to read than tAUS images due to the shadow of the pubic bone hiding the inferior end (far right) of the prostate from view (see Fig. 2b). While there tended to be good correlation for most measurements across all models, the RMSE errors were unacceptably high (particularly for the volume measurement). The discrepancy between the correlation results and the RMSE results can be partially explained by the method in which the SI dimension was determined. Even an inferred bounding box with an IoU of 1 would still overestimate the SI dimension. Addressing this would require altering the prolate ellipsoid formula by assuming the SI dimension is a fraction of the hypotenuse of the inferred box or by using an oriented-object detection algorithm (which infers the orientation of the bounding as well as its corners). This mismatch between different PV estimation techniques has been reported previously[23]. However, as the clinical value of PSAD operates within a range, and PV constitutes only one component of the PSAD calculation, the effect of quantifiable errors can be minimised. The cross-validation study results suggest that all the models, except for the YOLO^PB model, could potentially be used in calculating a volume measurement that correlates well with the volume calculated using the clinical approach of applying the prolate ellipsoid formula.

The results of the BA analysis showed overestimation of the volume measurement across all models before alteration of the SI dimension with LoAs ranges that would suggest the models cannot be used for volume estimation reliably. As with the RMSE volume values this was not unexpected due to the overestimation of the SI dimension. By halving the inferred SI dimension, both the bias and the range of the LoAs were dramatically reduced, with more measurements falling within the acceptable limits of $\pm 15 cm^3$ of the bias. While this may not be a clinically acceptable method of addressing the large range of the LoAs, it does suggest that simple alterations to the clinical approach to account for model measurement errors could result in acceptable volume errors using the models presented in this study.

When it comes to inference speed, YOLO is far superior to either RetinaNet or FasterRCNN. As ultrasound systems typically operate at about $20 - 30 fps$, only YOLO could be used in a truly real-time system, with RetinaNet and FasterRCNN requiring substantial frame rate limits in place.

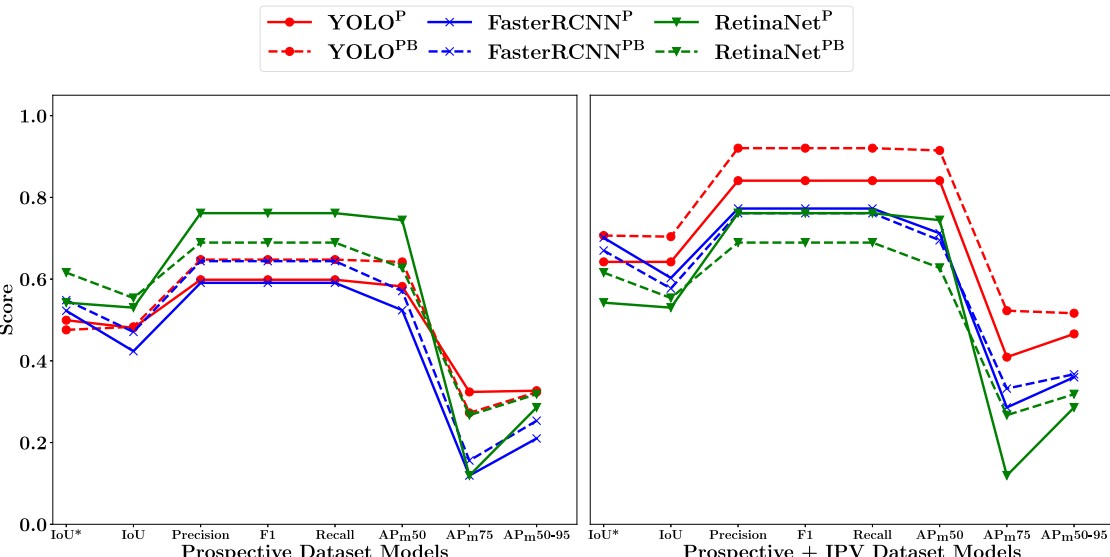

**Fig. 7 | Summary of prospective and IPV (image patch-voting) model metrics.** Left – prospective; Right – IPV. Dashed lines correspond to models trained to detect both bladder and prostate (superscript PB), with solid lines corresponding to models that were trained to detect prostate only (superscript P). IoU* (Intersection-over-union of highest confidence prostate bounding box); IoU (Intersection-over-union of all detected prostate boxes); AP (Average precision at varying thresholds).

While there are methods available for improving inference speeds for both RetinaNet and FasterRCNN models (pruning[24], quantisation[25], changing the backbone[26], etc.), the out-the-box models are substantially slower than YOLO. A preliminary test using FasterRCNN with a MobileNet[27] backbone was conducted, and it was found to substantially improve inference speed (from 137ms down to 58ms), with the effect on metric performance still to be analysed. YOLO also grants access to various model sizes, the smallest of which is only 6MB, and has an inference speed of about 15ms (67fps). This reduction in model size is usually accompanied by a drop in model performance, however, the results presented in this work warrant further study into the utility of various YOLO model sizes in prostate detection.

## Retrospective study

There was a noticeable drop in performance between the cross-validation study results and the results of the prospective models on the retrospective dataset. However, the prospective models were still able to identify the prostate and bladder relatively accurately, as indicated by the IoU* metric (all above 0.4). This drop in performance was largely rectified with the IPV models as the IPV dataset was 15 times larger than the prospective dataset, allowing the models to learn substantially more features, and return results more in line with the cross-validation study. The YOLO models benefitted the most from the addition of the IPV dataset, especially the YOLO^PB model, which performed the best across all metrics. The better performance of the YOLO models may be due to a more optimised training pipeline, which is part of the YOLO framework. The pipeline developed for the FasterRCNN and RetinaNet models was created to only mimic that of the YOLO pipeline, and a more in-depth analysis may show that what is optimal for YOLO is less than optimal for other network architectures.

In contrast to the cross-validation study, the SI dimension showed higher correlation than the AP dimension (when taken as an average across all models) but was still lower than the RL and volume measurement correlations. The addition of the IPV dataset did not have as drastic an effect as it did with the performance metrics, however, the average SI, AP, and volume measurement correlations still increased, and the average RL correlation decreased only slightly. The correlation spread between the models also reduced for the IPV models. As the volume measurement is of more importance, and all IPV models showed a correlation greater than 0.78, it is reasonable to assume that the IPV models could be used to calculate an initial estimate of the PV with a good correlation to expert manual estimates (provided an alteration of Equation (5) is made). Even though the RMSEs suggest otherwise, with volume RMSEs as high as $83.81cm^3$ for the IPV RetinaNet^P model, alterations to the prolate ellipsoid formula or using oriented-bounding box detection could help reduce the RMSEs to more acceptable levels.

The results of the prospective and IPV models on the special cases confirm that – using the models presented in this study – patients who have undergone benign interventions will need to be treated separately. However, it is possible that increasing the training dataset size, being careful to include more special case patients, could result in models that can handle non-special cases as well as special cases.

## Conclusions

Three SOTA object detection algorithms were trained and tested on multiple clinical datasets. These models were trained to detect the prostate and bladder in SUS images (particularly AUS images) of the prostate. All three models performed exceptionally well in all tests with minimal training optimisations applied. These results show that real-time tracking of the prostate using SUS scans is possible to within a relatively high degree of accuracy using current SOTA object detection algorithms. This opens the door to the development of a real-time SUS prostate detection tool that can aid probe operators during scanning of the prostate, ensuring high quality scans with minimal training. This in turn would make SUS a more reliable, repeatable, and clinically acceptable tool for measuring PV as an alternative option to invasive TRUS and certainly much cheaper than MRI.

Such a tool could make PCa screening/risk-stratification more accessible as it removes the need for a highly skilled clinician during scanning in the diagnostic pre-triaging and work-up of PCa while reducing patient discomfort. Although the rudimentary volume calculations returned high RMSEs, it was shown that minor alterations to the prolate ellipsoid formula could reduce BA LoAs to more acceptable ranges. This could enable volume estimation alongside real-time tracking, which could serve as a lower cost alternative for calculation of PSAD for the purposes of informing and prioritising clinical decision making in real-time.

As all models performed about the same (in the IoU* metric), when it comes to real-time analysis, YOLO's higher inference speeds make it more suitable for real-time tracking of the prostate.

The next steps include improving/optimising the RetinaNet and FasterRCNN network training processes; addressing the lower PCCs for the

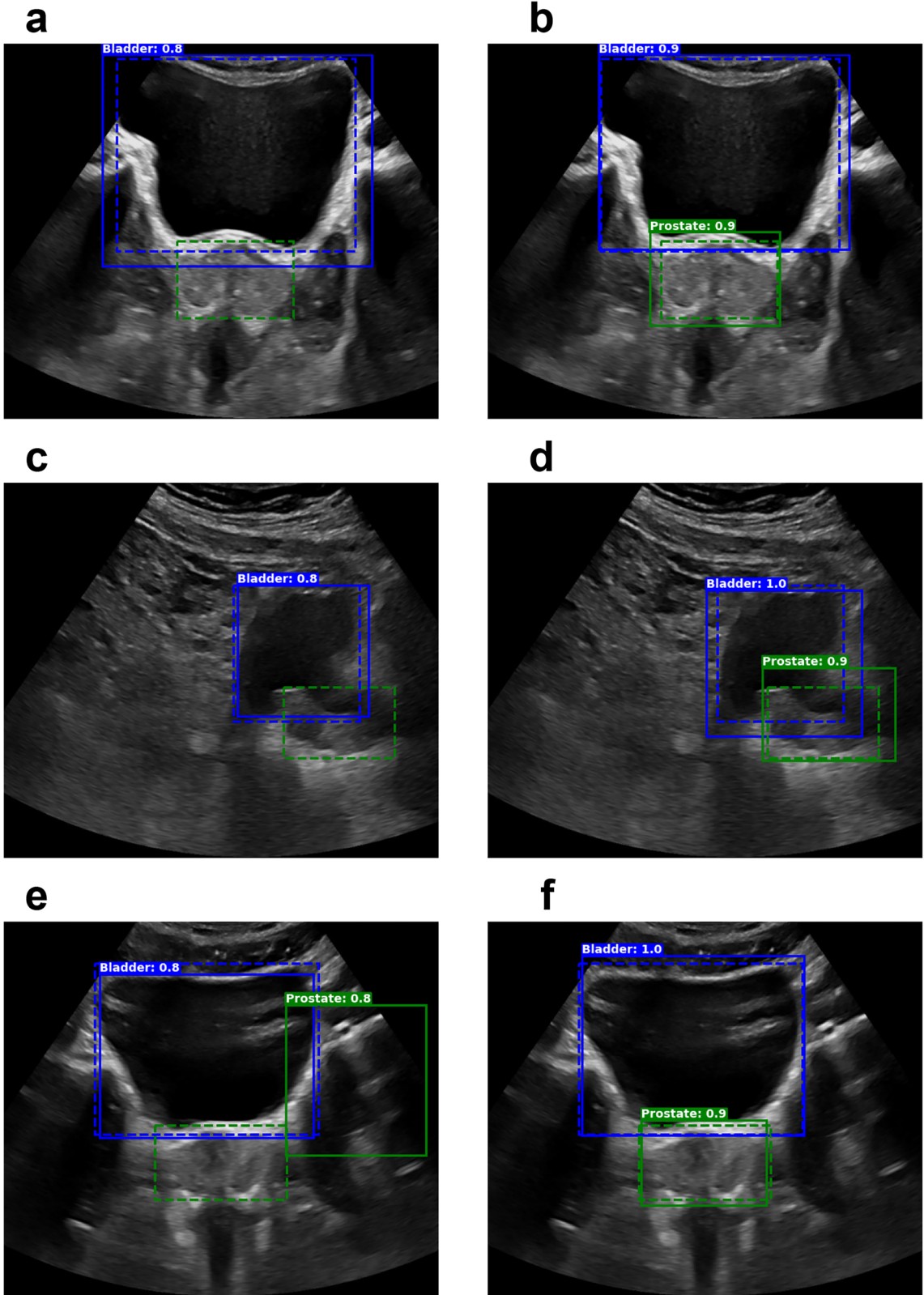

**Fig. 8 | Sample results highlighting the improvement between the prospective YOLO^PB model and the IPV (image patch-voting) YOLO^PB model. YOLO^PB – Left column; IPV YOLO^PB – Right column.** Each row corresponds to results from the same patient: **a**, **b**; **c**, **d**; and **e**, **f**. Model inferred boxes are solid lines, while ground truth boxes are dashed. Only the highest confidence boxes are shown, with green for prostate and blue for bladder. White text indicates box confidence. A missing solid green box ($P = 0.0$) indicates when the YOLO^PB model failed to identify the prostate at all.

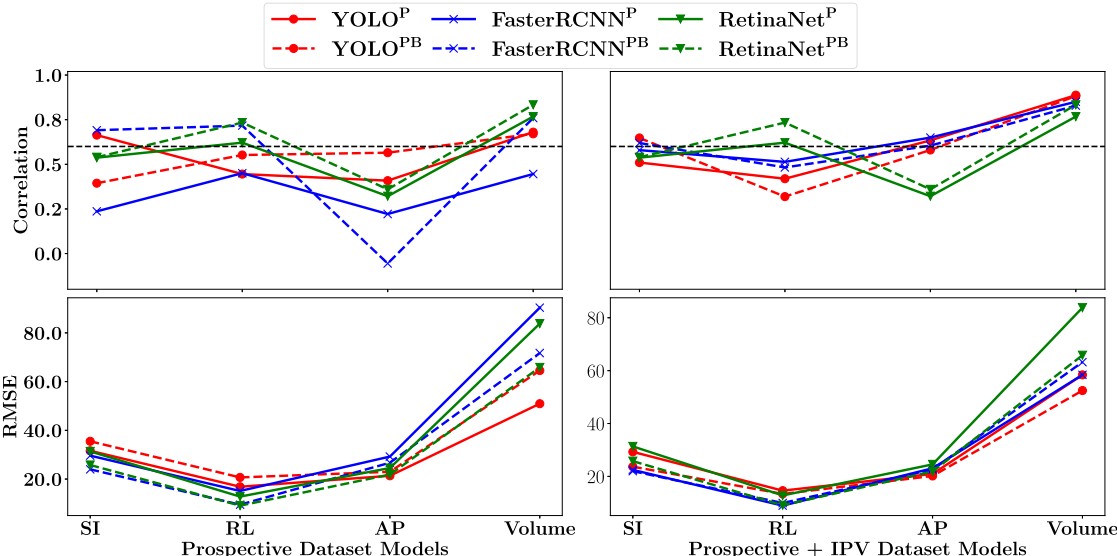

**Fig. 9 | Summary of prospective and IPV (image patch-voting) model PCCs (Pearson correlation coefficients) and RMSEs (root-mean-squared-errors) comparing ground truth dimensions with bounding box inferred dimensions for the YOLO (red), RetinaNet (green), and FasterRCNN (blue) models.** Prospective – Left column; IPV – Right column; Correlation – Top; RMSE – Bottom. The dashed lines correspond to models trained to detect both bladder and prostate (superscript PB), with solid lines corresponding to models that were trained to detect prostate only (superscript P). Only the highest confidence bounding box was used for the inferred dimension calculations. The black lines indicate a PCC of 0.6. SI (superior-inferior), RL (right-left), and AP (anterior-posterior) RMSE values are in *mm* (millimetres) while the Volume RMSE value is in $cm^3$ (cubic centimetres).

individual dimensions using oriented-bounding-box detection; testing the effect on performance of using smaller YOLO models; test the efficacy of oriented-bounding box detection models; increasing inference speed of the RetinaNet and FasterRCNN networks by using a backbone that is optimised for speed; and replacing the rudimentary volume calculation with a more sophisticated method using semantic segmentation. This work will culminate in the development of a clinical system to aid in the prostate scanning process using bounding boxes as guides and segmentation algorithms for PV inference.

## Methods

### Datasets

Three datasets were used in this study. The first two datasets (prospective and retrospective) had a combined total of 47 patients who were all scanned by the same experienced uro-radiologist. A Canon Aplio i700 ultrasound system using a curved linear array ultrasound transducer with a centre frequency of 3.5 *MHz* and a range of 1.9 *MHz* - 6 *MHz* was used. Patients in the prospective dataset were asked to present with a full bladder (ensuring the clearest view possible of the prostate when scanning) with the uro-radiologist performing the following scans: transabdominal (AUS) sweeps (transverse and sagittal) and transperineal sweeps (transverse and sagittal). To ensure a full bladder the patients were requested to drink a substantial amount of fluid at least 30 minutes prior to presenting for scanning. Written informed consent was obtained for all participants (ethics reference NRES 03/018). The retrospective dataset was collected for a previous study[23] where patient scans were collected retrospectively. The third dataset was taken from a publicly available dataset[21], referred to as the image-patch voting (IPV) dataset, consisting of 305 patients.

### Prospective Dataset

The prospective dataset was made up of 19 patients who had been referred for suspected PCa. 18 of these patients were under active surveillance with one patient being imaged prior to subsequent prostate biopsy. The median age of the patients was 75 years, ranging from 60 to 82 years old. For the present study, only the middle frames of the two AUS scans were used. The same experienced uro-radiologist then marked the prostate boundary of the middle frame of the prostate for each of the AUS scans.

### Retrospective Dataset

The retrospective dataset was made up of 28 patients who had undergone an AUS scan of the prostate within the preceding 12 months. This dataset was used exclusively to test the models that were trained on the data in the prospective and IPV datasets. It was decided that the retrospective dataset not be used for training purposes as some of the patients (6 in total) had undergone benign interventions such as transurethral resection of the prostate (TURP), patients which could be considered special cases when presenting for testing. Figure 2a, b shows examples of a non-special case that is subject to pubic bone shadowing and a special case where a catheter had been inserted into the patient's bladder, respectively.

### IPV dataset

The IPV dataset was collected and used by ref. 21 to train a network that calculated the size of the prostate following the clinical approach using the prolate ellipsoid calculation. Their dataset is made up of a single tAUS and single sAUS image of the centre of the prostate of 305 patients, with major dimensions of the prostate given. To ensure the bladder was always on the left and slightly above the prostate in the sagittal plane, minor preprocessing was applied in the current study. Images were flipped to ensure the bladder was always on the left and the prostate was on the right. The major dimension endpoints were then used to create bounding boxes that were used for training purposes.

### Models

5-fold cross-validation was employed to validate the different algorithms' utility in prostate and bladder detection on the prospective dataset only. Training and validation datasets were split by patient, where tAUS and sAUS images from the same patient were used for training or validating. Next, the entire prospective dataset was used to train models that were tested on the retrospective dataset. Finally, a set of models was trained on the combined prospective and IPV datasets, which were then tested on the retrospective dataset. The models trained on the full prospective dataset are referred to as the prospective models, and the models trained on the full prospective dataset combined with the IPV dataset are referred to as the IPV models. Models were trained to detect the prostate only (superscript of 'P') or the prostate and the bladder (superscript 'PB').

YOLO models were trained first due to the extensive online support offered. A portion of the RetinaNet and FasterRCNN models' training parameters were then set to mimic those used by YOLO. While YOLO comes as a complete package in Python, RetinaNet and FasterRCNN were built using PyTorch as the base package. Training and testing hardware specifications can be found in Supplementary Table 1. The selection of training parameter values, particularly those pertaining to data augmentation, was primarily based on what constituted reasonable transformations. For example, a random rotation of up to ± 30 degrees was deemed sufficient to introduce variability in the training data without positioning the prostate in an anatomically implausible location from a clinical imaging perspective. However, it should be noted that this parameter selection process was not formally optimised and represents an area for potential refinement in future work.

The preliminary tests with YOLO showed that training a single model using both the tAUS and sAUS images resulted in better performance in comparison to separate models trained on individual planes, therefore, only the multiplane models' results are presented in this paper. An important caveat to this is that the prospective dataset, used for the 5-fold cross-validation tests, was quite small. Larger datasets may show that separating models by plane returns better results.

### YOLO
Supplementary Table 2 contains the parameter values used during training of the YOLO models. The largest available model size for the YOLO models was used (roughly $120MB$ compared to RetinaNet's and FasterRCNN's sizes of $140MB$ and $165MB$, respectively). The maximum number of training epochs was set to 1000 with an early stopping patience of 100 epochs. By default, YOLO assigns a weight of 7.5 and 0.5 to the box loss and the classification loss, respectively. Images were resized to 600 pixels while maintaining the original images' aspect ratios.

### RetinaNet
Supplementary Table 2 also contains the parameter values used during training of the RetinaNet and FasterRCNN models. The RetinaNet models were built using PyTorch as the base package with the ResNet-50-FPN backbone (version 2)[28]. No pretrained weights were used. Stochastic gradient descent was chosen as the optimiser, and cosine annealing with warm restarts was used as the learning rate scheduler. A cosine learning rate was chosen to match the learning rate during training of the YOLO models. The maximum number of training epochs was set to 1000 with an early stopping patience of 100 epochs, and equal weighting was applied to all individual losses calculated during training. Oversampling was used with random transformations applied to the oversampled data only. Images were resized to 600 pixels while maintaining the original images' aspect ratios. Normalisation was applied, where the normalisation coefficients of the training data were calculated and applied to the training and validation data.

### FasterRCNN
The FasterRCNN model was set up in the same manner as the RetinaNet model with the ResNet-50-FPN backbone (version 2)[29]. All other parameters, including optimiser, learning rate scheduler, and transformations, were the same.

### Performance metrics
The intersection-over-union (IoU), average precision ($AP_m$), F1 score, precision, recall, and inference speed were analysed. To ensure the same metrics were being used, custom functions were written in Python instead of the built-in metrics made available for the YOLO model. While half of the models detected both prostate and bladder bounding boxes, metrics were only calculated using the prostate bounding boxes. For all models, a confidence threshold of 0.3 was used, where boxes detected with a confidence below the threshold were discarded immediately. In addition to the minimum confidence threshold, non-max suppression was applied with an IoU threshold of 0.5.

The calculation of IoU is given in Equation (1) where $A$ and $B$ are ground truth and inferred bounding boxes, respectively, and $\cap$ and $\cup$ are the intersection and union, respectively. An IoU of one indicates a 100% overlap between the inferred bounding box and the ground truth, while an IoU of zero indicates no overlap. The overall IoU is taken as the average of all inferred bounding boxes' IoUs (provided the inferred bounding boxes had a confidence above the confidence threshold). Since at most one prostate (or bladder) should be present in an image, a special IoU metric was used: IoU*, which indicates the IoU of the bounding box with the highest confidence.

$$IoU = \frac{A \cap B}{A \cup B} \tag{1}$$

Precision and Recall are calculated at a predetermined IoU and are given in Equation (2) where $TP$ is true positives, $FP$ is false positives, and $FN$ is false negatives (taken from the confusion matrix). $FN$ was taken as a detected bounding box with an IoU below a threshold of 0.5.

$$Precision = \frac{TP}{TP+FP}$$
$$Recall = \frac{TP}{TP+FN} \tag{2}$$

The F1 score is calculated as the harmonic mean between Precision and Recall and is given in Equation (3).

$$F1 = 2\frac{Precision \times Recall}{Precision + Recall} \tag{3}$$

$AP_m$ is calculated as the area under the Precision-Recall curve, given in Equation (4), where $p(r)$ is the equation of the Precision-Recall line. $AP_m50$ and $AP_m75$ have $IoU = 0.5$ and $IoU = 0.75$ as thresholds, respectively, and the $AP_m50$-95 metric is calculated as an average over the IoU range from 0.5 to 0.95 in steps of 0.05.

$$AP_m = \int_{r=0}^{1} p(r)dr \tag{4}$$

Inference speed was calculated as the time taken to pass the image through the relevant network during inference. The time taken to load the model into memory was not included, with bounding boxes inferred on a single image at a time (not in batches).

### Prostate size estimation
A rudimentary prostate size calculation, following the clinical workflow using the prolate ellipsoid formulation, was done. This involved estimating the three major dimensions of the prostate which were then used in the prolate ellipsoid calculation of Equation (5) where RL is the right-left dimension, AP is the anterior-posterior dimension, SI is the superior-inferior dimension, and C is a constant commonly chosen as $\frac{\pi}{6} \approx 0.52$ for the prolate ellipsoid assumption.

$$PV = RL \times AP \times SI \times C \tag{5}$$

Individual dimensions (RL, AP, and SI) were compared with their respective ground truths, as was the calculated PV. The RL and AP dimensions were taken from the bounding box inferred on the tAUS images, and the SI dimension was taken from the bounding box on the sAUS images, as shown in Fig. 10. All inferred dimensions and final volumes were compared to their respective ground truths using the Pearson's Correlation Coefficient (PCC) as well as root-mean-squared-error (RMSE) value. A PCC of 0.6 was taken as high correlation. Volume Bland-Altman (BA) analyses were carried out for the cross-validation study with the plots given in the Supplementary Figs. 1–3. Differences calculated in $cm^3$ and as error percentages were considered. Due to the oversimplified acquisition of the SI dimension large limits-of-agreement (LoA) were anticipated, and only a brief discussion of the BA plots is given in the Results section.

**Fig. 10 | Conversion of bounding boxes into prolate ellipsoid dimensions for the calculation of PV (prostate volume) using the clinical workflow.** RL (right-left) and AP (anterior-posterior) dimensions from a tAUS (transverse transabdominal ultrasound) image (**a**), SI (superior-inferior) dimension from a sAUS (sagittal transabdominal ultrasound) image (**b**), and prolate ellipsoid used in the clinical workflow **c**.

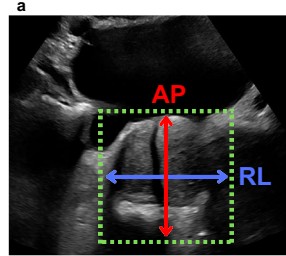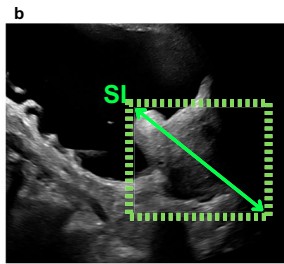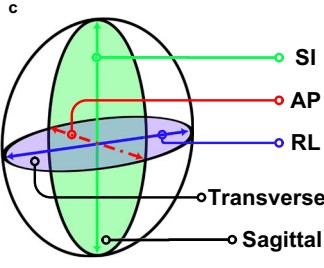

## Ethics approval and consent to participate
All elements of this prospective study were carried out in accordance with the Declaration of Helsinki and were approved by the institutional ethics board (NRES Committee East of England, UK, ref: NRES 03/018), with written informed consent obtained from all participants. All methods were performed in accordance with the relevant guidelines and regulations.

## Reporting summary
Further information on research design is available in the Nature Portfolio Reporting Summary linked to this article.

## Data availability
Data is provided within the manuscript or supplementary information files.

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

## Acknowledgements

This research utilised Queen Mary's Apocrita HPC facility, supported by QMUL Research-IT: https://doi.org/10.5281/zenodo.438045. Vincent J. Gnanapragasam acknowledges infrastructure support from the UK National Institute for Health Research (NIHR) Cambridge Biomedical Research Centre (BRC-1215-20014). The views expressed are those of the authors and not necessarily those of the NIHR or the Department of Health and Social Care. Tristan Barrett acknowledges support from the NIHR Cambridge Biomedical Research Centre (NIHR203312) and Cancer Research UK (Cambridge Imaging Centre grant number C197/A16465), the Engineering and Physical Sciences Research Council Imaging Centre in Cambridge and Manchester, and the Cambridge Experimental Cancer Medicine Centre. The views expressed are those of the authors and not necessarily those of the NIHR or the Department of Health and Social Care. Zion Tsz Ho Tse acknowledges funding support from the Academy of Medical Sciences Professorship, Royal Society Wolfson Fellowship, Cancer Research UK (EDDPMA-Nov21\100026), and National Institutes of Health (NIH) Bench-to-Bedside Award. This study also received support from the NIH Center for Interventional Oncology: Grant ZID# BC011242 & CL040015, and the Intramural Research Program of the National Institutes of Health.

## Author contributions

R.D.B. wrote the main manuscript text. Z.T.H.T., V. J. G., and T.B. provided intelligent ideas, the funding, and edited and revised the manuscript. All authors reviewed the manuscript.

## Competing interests

The authors declare no competing interests.
