## [Transparent Peer Review file · Communications Engineering]

Object Detection as an Aid for Locating the Prostate in Surface-Based Abdominal Ultrasound Images

Corresponding Author: Professor Zion Tse

Version 0:

Reviewer comments:

Reviewer #1

(Remarks to the Author)

The claim of the paper is to show that prostate detection from surface-based ultrasound images can be done using state-of-the-art deep learning object detection algorithms in real time. The study also claims that the object detection methods can be used to estimate prostate volume with a high correlation to ground truth values. The study uses YOLOv8, RetinaNet and FasterRCNN models for the detection of prostate.

The novelty of the study can be summarized as the utilization of state-of-the-art models for prostate detection from surface-based ultrasound images and presenting a real-time method. The study is valuable for the field analyzing the state-of-the-art models to detect the prostate with and without the bladder from multiplane images. The study will be a pioneer for new studies in the field.

Although the study has novel sides, the statement "... to the best of the authors knowledge, deep learning guided prostate detection using surface-based ultrasound images has not been investigated before" does not reflect the truth. The 17th reference, which is the source of one of the datasets used by the study, is one of our previous studies and presents an end-to-end system for the estimation of prostate volume from abdominal ultrasound images. The end-to-end system involves both the detection of the prostate and the estimation of the prostate volume. The authors of the 17th reference are also given wrong.

The paper should be revised by correcting the errors mentioned above.

Reviewer #2

(Remarks to the Author)

This manuscript is focused on showcasing various AI object detection models for identifying prostates and estimating their dimensions and overall volume. Overall, the manuscript is well-organized and detailed, however there are some aspects that could be improved as detailed below:

Introduction: while prostate ultrasound measurements are novel, there is a breadth of background material related to ultrasound and AI that could be part of the literature review for this manuscript. This would help the reader know how this fits with other applications and better support the justification for the three object detection models used. Why YOLO v8 instead of the numerous other versions of YOLO?

Datasets: Wording is confusing to state the "first dataset" and in the same paragraph discuss the third dataset without mention explicitly of second dataset. Recommend to more explicitly separate mention of first two datasets vs. saying subdivided from the one.

Datasets: How is full bladder defined as used in the first and second datasets? Figure 1A and 1B have very different size bladders so curious how this is accounted for.

Datasets: The authors state this is the first manuscript to use AI for prostate volume estimates from SUS images. How does this work differ from [17] as the authors use the dataset from this paper where a network was trained to calculate the size of

the prostate per the dataset section. How does the QDCNN model from [17] differ from this work? More details should be added in the introductory remarks.

Methods: Is the 5-fold cross validation across the entire dataset or split by subjects?

Methods: Was the retrospective dataset used for model testing include the “special cases” or were those removed from the test dataset. These cases were mentioned as not wanting to be included in the training data so it seems odd to use them then to measure model performance. Clarification or removal of these data is recommended.

Methods: Were the 1000 epochs used for RetinaNet and FasterRCNN? Were the specific training parameters for each model arbitrary, default, or was some justification used? The reasoning should be stated in the paper.

Methods: Image augmentation parameters were shown in the supplement but not mentioned in manuscript. Why would different augmentation approaches be used for each model and how does that impact the results?

Methods: Do the 120, 140 and 165 MB sizes include the default training weights that were not used in this study?

Methods: The differences in IoU and IoU* are confusing and need more explanation. Are the authors saying all confidence predictions were used for IoU with no confidence cut-offs, such as all the predictions in Figure 8A?

Methods: What is the 0.52 constant?

Methods: It is confusing to have a performance metric AP and a prostate dimension AP. Could one of these be renamed or given a subscript indicator?

Results: Is there a reason or discussion from the authors as to why the prostate only models seem to outperform the prostate-bladder models? Discussion on this trend would help the manuscript.

Results: Table of metrics would help vs. Figure 3. Also it is stated RetinaNet and FasterRCNN have higher confidence values vs. YOLO – is this data available as it should be added to the manuscript if it is.

Results: Figures and words used different notation for model names and should be consistent (YOLO^{PB} vs yolo_pb)

Results: What is the “R” in the ultrasound images – should state in figure caption.

Results: Figure 5, correlation values are not sufficient for showing how well data tracks ground truths. Root mean square error values or other error measurements for each dimensions and volume to quantify how off these measurement are from the ground truth.

Discussion: New results such as those shown in the figures should be first introduced in the results section.

Discussion: More discussion on how the results overall compare to the ground truth would be useful. What is the acceptable error rate for these measurement clinically? What is the error rate for the AI predictions? It would help to frame things in this context to better understand how close the introduced AI models are to being able to supplement clinical care.

Reviewer #3

(Remarks to the Author)

Summary: This paper aims for automatic prostate localization in ultrasound using SOTA detection models such as Yolo, RetinaNet, and FasterRCNN. The evaluations on three datasets demonstrates the models’ high accuracy in detection and estimation of prostate volume across multiple performance metrics. The results also demonstrate the accuracy for the real-time tracking of the prostate using SOTA detection models. In terms of weakness, the paper lacks several critical discussion and evaluations that would strengthen its novelty and clinical relevance.

This paper shows limited novelty beyond technical implementation of known models - Object detection is a well-investigated area in medical imaging even for prostate localization. Adapting existing technical models (Yolo, FasterRCNN etc.) to ultrasound imagery may lack sufficient novelty. It also lacks a thorough discussion of the clinical workflow integration and decision support implications.

The paper lacks discussion and justification with existing research related - There are already studies leveraging object detection for prostate localization and volume estimation, including Yolo-based methods. For example, related work like[1] could be discussed and critically compared.

The authors claim the potential of using technical models to enhance clinical workflows. It is also suggested to include clinical evaluation or user study to support this claim. Showing how the tool affects e.g. clinicial decision-making and time for scanning would strengthen its practical relevance.

For experiments, the paper reports across several performance metrics (IoU, percision, Pearson correlation coefficient, etc.).

It lacks statistical analysis to determine whether differences across models or datasets are significant. E.g., Bland–Altman (BA) plots can be used and analysed to assess agreement of models beyond correlation.

[1] Automated prostate cancer grading and diagnosis system using deep learning-based YOLO object detection algorithm

Version 1:

Reviewer comments:

Reviewer #1

(Remarks to the Author)

I appreciate the authors' revisions, which have substantially improved the manuscript. The authors have adequately addressed the reviewers' concerns. I believe the manuscript is suitable for publication.

Reviewer #2

(Remarks to the Author)

Thanks for taking the time to respond to the queries. The authors have sufficiently addressed the comments.

Reviewer #3

(Remarks to the Author)

The response letter has adequately addressed my concerns. However, the figures that are currently included in the supplementary material would substantially enhance the clarity and impact of the work if moved to the main manuscript. Incorporating these figures into the primary text will make the results more accessible to readers and strengthen the overall presentation.

Reviewer Comments

We would like to thank you and the reviewers for taking the time to evaluate our manuscript and provide detailed and thoughtful feedback. We believe that most of the comments can be addressed in a manner that will strengthen the manuscript overall. We remain enthusiastic about the possibility of publishing with your journal and acknowledge the reviewers' concerns regarding the novelty of our work.

Line numbers and page numbers have been added to the manuscript to help with locating concerns.

The marked up document highlights the major changes made to the manuscript.

Reviewer 1

1. **Comment:** The claim of the paper is to show that prostate detection from surface-based ultrasound images can be done using state-of-the-art deep learning object detection algorithms in real time. The study also claims that the object detection methods can be used to estimate prostate volume with a high correlation to ground truth values. The study uses YOLOv8, RetinaNet and FasterRCNN models for the detection of prostate.

Response: No changes made.

2. **Comment:** The novelty of the study can be summarized as the utilization of state-of-the-art models for prostate detection from surface-based ultrasound images and presenting a real-time method. The study is valuable for the field analyzing the state-of-the-art models to detect the prostate with and without the bladder from multiplane images. The study will be a pioneer for new studies in the field.

Response: No changes made.

3. **Comment:** Although the study has novel sides, the statement "... to the best of the authors knowledge, deep learning guided prostate detection using surface-based ultrasound images has not been investigated before" does not reflect the truth. The 17th reference, which is the source of one of the datasets used by the study, is one of our previous studies and presents an end-to-end system for the estimation of prostate volume from abdominal ultrasound images. The end-to-end system involves both the detection of the prostate and the estimation of the prostate volume. The authors of the 17th reference are also given wrong.

Response:

- a. The work presented in the reference (now reference 21) is landmark (key-point) identification. While the landmarks do belong to the prostate, and therefore one can assume that if the landmarks are detected in the image then the prostate must be present, it is not quite the same as detecting the prostate as a classed object in an image. The introduction has been expanded to highlight the difference between object detection and landmark identification, with more details of the reference given. **Lines 58 – 68.**
 - b. Incorrect authorship noted and corrected (Mendeley added incorrect authors to reference).
4. **Comment:** The paper should be revised by correcting the errors mentioned above.

Response: No changes made.

Reviewer 2

1. **Comment:** This manuscript is focused on showcasing various AI object detection models for identifying prostates and estimating their dimensions and overall volume. Overall, the manuscript is well-organized and detailed, however there are some aspects that could be improved as detailed below.

Response: No changes made.

2. **Comment:** Introduction: while prostate ultrasound measurements are novel, there is a breadth of background material related to ultrasound and AI that could be part of the literature review for this manuscript. This would help the reader know how this fits with other applications and better support the justification for the three object detection models used. Why YOLO v8 instead of the numerous other versions of YOLO?

Response:

- a. 4 references were included. These 4 references were chosen to highlight how YOLO, FasterRCNN, and RetinaNet have been used for real-time (or near real-time) object detection across various fields. One of the references specifically looked at YOLO in the medical field. The choice for the three models came down to speed. FasterRCNN, RetinaNet, and YOLO are known for the faster inference times, and while not all of them turned out to be truly real time capable with the hardware used, they returned results in a comparable manner. This should be clearer now in **Lines 39 – 48**.
- b. V8 was the latest version when this work started. This has been clarified in the background. **Line 40**.

3. **Comment:** Datasets: Wording is confusing to state the “first dataset” and in the same paragraph discuss the third dataset without mention explicitly of second dataset. Recommend to more explicitly separate mention of first two datasets vs. saying subdivided from the one.

Response: The ambiguity regarding dataset names and descriptions is noted. There are 3 datasets: prospective dataset (19 patients), retrospective dataset (28 patients), and IPV dataset (305 patients). The prospective and retrospective datasets were collected by the same uro-radiologist at different dates, with the prospective dataset collected specifically for this study and the retrospective dataset collected for a previous study. The opening paragraph of the Datasets section has been updated and should highlight the differences in the datasets more clearly. **Lines 75 – 84**.

4. **Comment:** Datasets: How is full bladder defined as used in the first and second datasets? Figure 1A and 1B have very different size bladders so curious how this is accounted for.

Response: A “full” bladder was asking the patients to have a substantial drink at least 30 minutes before presenting for the scanning session. Figure 1A and 1B were specifically chosen to highlight that the retrospective dataset contained images that were different from the norm. 1B shows a patient with a catheter inserted into their bladder, which explains the substantial difference in bladder sizes. This has been clarified in the first Datasets paragraph. **Lines 80 - 81**.

5. **Comment:** Datasets: The authors state this is the first manuscript to use AI for prostate volume estimates from SUS images. How does this work differ from [17] as the authors use the dataset from this paper where a network was trained to calculate the size of the prostate per the dataset section. How does the QDCNN model from [17] differ from this work? More details should be added in the introductory remarks.

Response: The claim “Object detection is a well-researched field across myriad domains, however, to the best of the authors knowledge, deep learning guided prostate detection using SUS images has not been investigated before” still holds. As mentioned for reviewer 1 comment 3, reference 17 (now 21) conducted landmark identification and not object detection. The introduction has been

expanded to highlight the difference between object detection and landmark identification. **Lines 58 – 68.**

6. **Comment:** Methods: Is the 5-fold cross validation across the entire dataset or split by subjects?
Response: Ambiguity noted. The cross-validation was split by subject/patient, e.g. fold 0 was trained using data from patients 1, 2, 3, 5, 7, 8, 9, 10, 11, 12, 14, 15, 16, 17, 19 and validated on data from patients 4, 6, 13, 18. This should be clearer with the added statement. **Lines 108 – 109.**

7. **Comment:** Methods: Was the retrospective dataset used for model testing include the “special cases” or were those removed from the test dataset. These cases were mentioned as not wanting to be included in the training data so it seems odd to use them then to measure model performance. Clarification or removal of these data is recommended.
Response: During testing on the retrospective dataset the special cases were included. This was done as only a minor performance drop was noted when including the special cases. To aid in clarification, the special cases are now not included in the test dataset and have their own section. The changes now highlight how the models perform when presented with irregular prostate images, with examples given in the Supplemental Materials. The Retrospective Results section has been rewritten with this in mind. **Lines 229 – 257.**

8. **Comment:** Methods: Were the 1000 epochs used for RetinaNet and FasterRCNN? Were the specific training parameters for each model arbitrary, default, or was some justification used? The reasoning should be stated in the paper.
Response:
 - a. Clarification: 1000 epochs was the maximum allowable epochs for all models, however, the patience parameter of 100 epochs meant that in almost all cases training ended well before reaching the 1000 epochs limit as model improvement over 100 epochs was not recorded. **No changes made.**
 - b. The selection of training parameter values was based on what constituted reasonable transformations from a clinical imaging perspective. The transformations used by RetinaNet and FasterRCNN were selected to match those used by YOLO as closely as possible. The second paragraph of the Methods section has been updated to highlight this. **Lines 115 - 123.**

9. **Comment:** Methods: Image augmentation parameters were shown in the supplement but not mentioned in manuscript. Why would different augmentation approaches be used for each model and how does that impact the results?
Response: YOLO has “built-in” augmentation, where the user can set some parameter values during training. RetinaNet and FasterRCNN required augmentation pipelines to be created manually. RetinaNet and FasterRCNN made use of the exact same augmentation pipelines, which were designed to mimic YOLO as closely as possible, however, random seeding and instantiation make it a little difficult for them to be a perfect match. Rewording of parts of the Methods section should have made this clearer. **Lines 107 – 146.**

10. **Comment:** Methods: Do the 120, 140 and 165 MB sizes include the default training weights that were not used in this study?
Response: These are the sizes of the models once training is complete (and saved to disk), training that was done without pretrained weights. The number of weights is not affected by pretrained or not pretrained, only the starting values of the weights are. **No changes made.**

11. **Comment:** Methods: The differences in IoU and IoU* are confusing and need more explanation. Are the authors saying all confidence predictions were used for IoU with no confidence cut-offs, such as all the predictions in Figure 8A?
Response: The IoU calculations included all inferred bounding boxes returned by a model assuming a confidence threshold of 0.3 (boxes with a confidence below this threshold were immediately discarded). The box with the highest confidence of all inferred boxes (all above the confidence threshold) was used for the IoU* metric. This should be clearer now. The confidence threshold used by all models has been highlighted in **Lines 151-152**, with an expansion of the IoU and IoU* given in **Lines 158-159**.
12. **Comment:** Methods: What is the 0.52 constant?
Response: This is the constant used by clinicians in the ellipsoid assumption of the prostate volume calculation. A few more words have been added to clarify this value. **Line 174**.
13. **Comment:** Methods: It is confusing to have a performance metric AP and a prostate dimension AP. Could one of these be renamed or given a subscript indicator?
Response: The AP acronym for the metric has been changed to AP_m (AP metric) throughout the manuscript, including figures.
14. **Comment:** Results: Is there a reason or discussion from the authors as to why the prostate only models seem to outperform the prostate-bladder models? Discussion on this trend would help the manuscript.
Response: The exact reason is unknown. It could be due to multiclass models needing to learn more complex features. They no longer need to only detect the presence of an object or not, they need to distinguish between object types. The limited dataset size makes drawing any solid conclusions difficult, and this has been noted in the Discussion section. **Lines 266 – 271**.
15. **Comment:** Results: Table of metrics would help vs. Figure 3. Also it is stated RetinaNet and FasterRCNN have higher confidence values vs. YOLO – is this data available as it should be added to the manuscript if it is.
Response: Numerical tables have been added to the Supplemental Materials for all plots included in the main manuscript. The average confidence values have also been added to these tables for the 5-fold cross-validation study in the Supplemental Materials.
16. **Comment:** Results: Figures and words used different notation for model names and should be consistent (YOLO[^]PB vs yolo_pb)
Response: This has been addressed. Figures have been updated to match text with superscripts.
17. **Comment:** Results: What is the “R” in the ultrasound images – should state in figure caption.
Response: This was a watermark used purely for identification purposes. It has been removed from all images in the manuscript.
18. **Comment:** Results: Figure 5, correlation values are not sufficient for showing how well data tracks ground truths. Root mean square error values or other error measurements for each dimensions and volume to quantify how off these measurement are from the ground truth.
Response: RMSE values have been added for the cross-validation study and the retrospective study for all dimensions and volumes. There is a fundamental flaw in how the SI dimension is calculated: the bounding box hypotenuse will always be larger than the SI dimension, even if the box has an IoU of 1 (perfect match). This in turn results in a larger volume error. If dimensions calculated from bounding boxes were to be used in the volume calculation, a slightly altered prolate ellipsoid

formula would need to be developed. This is explained in more detail in the Discussion section with a discussion on the role PV plays in PSAD calculation. **Lines 276 – 287.**

19. **Comment:** Discussion: New results such as those shown in the figures should be first introduced in the results section.

Response: Results (images/figures) have been moved from the Discussion section and placed in their respective Results sections.

20. **Comment:** Discussion: More discussion on how the results overall compare to the ground truth would be useful. What is the acceptable error rate for these measurement clinically? What is the error rate for the AI predictions? It would help to frame things in this context to better understand how close the introduced AI models are to being able to supplement clinical care.

Response: The inclusion of the RMSEs should cover the comparison to the ground truth. The authors of reference 22 have reported that there is often a mismatch between different PV estimation techniques. This is not necessarily a problem as PSAD (as part of the diagnostic work-up to PCa) operates in a range, where clinical thresholds can be adjusted as necessary. The results presented were intended as a preliminary study to highlight that object detections algorithms can locate the prostate in SUS images. The development of a clinically viable system forms part of the next study currently underway. **No changes made.**

Reviewer 3

1. **Response:** Summary: This paper aims for automatic prostate localization in ultrasound using SOTA detection models such as Yolo, RetinaNet, and FasterRCNN. The evaluations on three datasets demonstrates the models' high accuracy in detection and estimation of prostate volume across multiple performance metrics. The results also demonstrate the accuracy for the real-time tracking of the prostate using SOTA detection models. In terms of weakness, the paper lacks several critical discussion and evaluations that would strengthen its novelty and clinical relevance.

Response: No changes made.

2. **Comment:** This paper shows limited novelty beyond technical implementation of known models - Object detection is a well-investigated area in medical imaging even for prostate localization. Adapting existing technical models (Yolo, FasterRCNN etc.) to ultrasound imagery may lack sufficient novelty. It also lacks a thorough discussion of the clinical workflow integration and decision support implications.

Response: The majority of prostate localisation is either for TRUS or MR images, not SUS images, which is included in the Background. The Background has also been expanded to highlight where the results of this study will be applied in a clinical setting. **Lines 49 – 57 with Figure 1.**

3. **Comment:** The paper lacks discussion and justification with existing research related - There are already studies leveraging object detection for prostate localization and volume estimation, including Yolo-based methods. For example, related work like [1] could be discussed and critically compared.

Response: Apart from the landmark identification reference (now 21), there are no studies that attempt prostate detection using SUS images. There are a multitude of studies that attempt object detection using MRI and/or TRUS images. Due to the lower image quality of SUS when compared to MRI and/or TRUS one would naturally expect MRI/TRUS based models to outperform SUS based models. However, the atraumatic nature and lower relative cost of SUS systems make them an attractive alternative for screening/risk-stratification/triage purposes. While YOLO was used in the suggested reference, they were detecting prostate cancer in histological images. The current manuscript does not detect prostate cancer, it detects the prostate as part of a workflow that will be used in risk-stratification/screening protocols, prior to detecting prostate cancer. **No changes made.**

4. **Comment:** The authors claim the potential of using technical models to enhance clinical workflows. It is also suggested to include clinical evaluation or user study to support this claim. Showing how the tool affects e.g. clinical decision-making and time for scanning would strengthen its practical relevance.

Response: The current study is a preliminary study into whether AI and machine learning can augment the current diagnostic work-up to PCa. The next steps will be to develop a system that can be validated clinically with a user study. **No changes made.**

5. **Comment:** For experiments, the paper reports across several performance metrics (IoU, percision, Pearson correlation coefficient, etc.). It lacks statistical analysis to determine whether differences across models or datasets are significant. E.g., Bland–Altman (BA) plots can be used and analysed to assess agreement of models beyond correlation.

Response: A BA analysis has been added. Due to the overestimation of the SI dimension a highly accurate volume measurement was not expected, hence the use of correlation instead of a difference analysis. The BA plots have been added to the Supplemental Materials, with a brief discussion added to the results section for the cross-validation study. **Lines 212 – 219.**

Reviewer Comments

Reviewer 1

1. **Comment:** The claim of the paper is to show that prostate detection from surface-based ultrasound images can be done using state-of-the-art deep learning object detection algorithms in real time. The study also claims that the object detection methods can be used to estimate prostate volume with a high correlation to ground truth values. The study uses YOLOv8, RetinaNet and FasterRCNN models for the detection of prostate.

Response: No changes necessary.

2. **Comment:** The novelty of the study can be summarized as the utilization of state-of-the-art models for prostate detection from surface-based ultrasound images and presenting a real-time method. The study is valuable for the field analyzing the state-of-the-art models to detect the prostate with and without the bladder from multiplane images. The study will be a pioneer for new studies in the field.

Response: No changes necessary.

3. **Comment:** Although the study has novel sides, the statement "... to the best of the authors knowledge, deep learning guided prostate detection using surface-based ultrasound images has not been investigated before" does not reflect the truth. The 17th reference, which is the source of one of the datasets used by the study, is one of our previous studies and presents an end-to-end system for the estimation of prostate volume from abdominal ultrasound images. The end-to-end system involves both the detection of the prostate and the estimation of the prostate volume. The authors of the 17th reference are also given wrong.

Response:

- a. The work presented in reference 17 is landmark (key-point) identification. While the landmarks do belong to the prostate, and therefor one can assume that if the landmarks are detected in the image, then the prostate must be present, it is not quite the as detecting the prostate as a generic object in an image. This could be clarified, either by expanding the current study to include a section that shows object detection can detect a lack of prostate, or by noting the difference between landmark identification and object detection.
- b. Incorrect authorship will need to be fixed.

4. **Comment:** The paper should be revised by correcting the errors mentioned above.

Response: No changes necessary.

Reviewer 2

1. **Comment:** This manuscript is focused on showcasing various AI object detection models for identifying prostates and estimating their dimensions and overall volume. Overall, the manuscript is well-organized and detailed, however there are some aspects that could be improved as detailed below.

Response: No changes necessary.

2. **Comment:** Introduction: while prostate ultrasound measurements are novel, there is a breadth of background material related to ultrasound and AI that could be part of the literature review for this manuscript. This would help the reader know how this fits with other applications and better support the justification for the three object detection models used. Why YOLO v8 instead of the numerous other versions of YOLO?

Response:

- a. Introduction can be expanded to include some TRUS prostate AI references to round it out. Show that AI has utility in prostate US. See if RetinaNet, FaterRCNN, or YOLO have been used here before.
- b. V8 was the latest version when this work started. There have been 4 updates since then, so keeping on top of the latest model is tough.

3. **Comment:** Datasets: Wording is confusing to state the “first dataset” and in the same paragraph discuss the third dataset without mention explicitly of second dataset. Recommend to more explicitly separate mention of first two datasets vs. saying subdivided from the one.

Response: Must clarify dataset nomenclature. This has been a recurring problem and it definitely still needs work.

4. **Comment:** Datasets: How is full bladder defined as used in the first and second datasets? Figure 1A and 1B have very different size bladders so curious how this is accounted for.

Response: A “full” bladder was asking the patients to have a drink before presenting for the scanning.

5. **Comment:** Datasets: The authors state this is the first manuscript to use AI for prostate volume estimates from SUS images. How does this work differ from [17] as the authors use the dataset from this paper where a network was trained to calculate the size of the prostate per the dataset section. How does the QDCNN model from [17] differ from this work? More details should be added in the introductory remarks.

Response: *The claim “Object detection is a well-researched field across myriad domains, however, to the best of the authors knowledge, deep learning guided prostate detection using SUS images has not been investigated before” still holds. As mentioned for reviewer 1 comment 3, reference 17 conducted landmark identification and not object detection. This can be clarified either in the introduction, or with a slight expansion of the study.*

6. **Comment:** Methods: Is the 5-fold cross validation across the entire dataset or split by subjects?

Response: Clarify this. Split by subjects.

7. **Comment:** Methods: Was the retrospective dataset used for model testing include the “special cases” or were those removed from the test dataset. These cases were mentioned as not wanting to be included in the training data so it seems odd to use them then to measure model performance. Clarification or removal of these data is recommended.

Response: Verify what was done.

8. **Comment:** Methods: Were the 1000 epochs used for RetinaNet and FasterRCNN? Were the specific training parameters for each model arbitrary, default, or was some justification used? The reasoning should be stated in the paper.

Response:

- a. Clarification: 1000 epochs was the maximum allowable epochs, however the patience parameter of 300 epochs meant that in almost all cases training ended well before 1000 epochs.
- b. For the training parameters, RetinaNet and FasterRCNN were altered in an attempt to match YOLO, but not perfectly. Image size, patience, learning rates, were a close match. RetinaNet and FasterRCNN required custom implementation of augmentation. Similar augmentations as YOLO were used, but again not exactly the same.
- c. Will need to clarify this in the paper.

9. **Comment:** Methods: Image augmentation parameters were shown in the supplement but not mentioned in manuscript. Why would different augmentation approaches be used for each model and how does that impact the results?

Response: YOLO has “built in” augmentation, where the user can set some parameter values during training. RetinaNet and FasterRCNN required that these augmentation pipelines be created manually. RetinaNet and FasterRCNN made use of the exact same augmentations, which were designed to mimic YOLO closely, but not perfectly. Random seeding and instantiation make it a little difficult for it to be a perfect match.

10. **Comment:** Methods: Do the 120, 140 and 165 MB sizes include the default training weights that were not used in this study?

Response: Clarify: These are the sizes of the models once training is complete, training that was done without pretrained weights. The number of weights is not affected by pretrained or not pretrained, only the starting value of the weights are.

11. **Comment:** Methods: The differences in IoU and IoU* are confusing and need more explanation. Are the authors saying all confidence predictions were used for IoU with no confidence cut-offs, such as all the predictions in Figure 8A?

Response: Clarify: The IoU calculations had a cut-off of 0.3 (default value), and 0.7 for NMS. The highest confidence of all boxes that still came through after the 0.3 conf cutoff and 0.7 nms was used for IoU*.

12. **Comment:** Methods: What is the 0.52 constant?

Response: Clarify: The constant used by clinicians in the ellipsoid assumption of the prostate volume calculation.

13. **Comment:** Methods: It is confusing to have a performance metric AP and a prostate dimension AP. Could one of these be renamed or given a subscript indicator?

Response: Will change the dimensions to AP_dim, RL_dim, SI_dim.

14. **Comment:** Results: Is there a reason or discussion from the authors as to why the prostate only models seem to outperform the prostate-bladder models? Discussion on this trend would help the manuscript.

Response: Will have a look at this.

15. **Comment:** Results: Table of metrics would help vs. Figure 3. Also it is stated RetinaNet and FasterRCNN have higher confidence values vs. YOLO – is this data available as it should be added to the manuscript if it is.

Response: It can be made available.

16. **Comment:** Results: Figures and words used different notation for model names and should be consistent (YOLO^PB vs yolo_pb)

Response: Must fix this.

17. **Comment:** Results: What is the “R” in the ultrasound images – should state in figure caption.

Response: That is my watermark, but I think we should have a lab watermark. This can be removed if necessary.

18. **Comment:** Results: Figure 5, correlation values are not sufficient for showing how well data tracks ground truths. Root mean square error values or other error measurements for each dimensions and volume to quantify how off these measurement are from the ground truth.

Response: This can be added quite easily. As well as a statistical analysis.

19. **Comment:** Discussion: New results such as those shown in the figures should be first introduced in the results section.

Response: Must have a look at this.

20. **Comment:** Discussion: More discussion on how the results overall compare to the ground truth would be useful. What is the acceptable error rate for these measurement clinically? What is the error rate for the AI predictions? It would help to frame things in this context to better understand how close the introduced AI models are to being able to supplement clinical care.

Response: The dimension/volume results were not supposed to be the focus of the paper, they were added as a quick aside as to what can be done with the detection results, hence the lack of comparison with ground truth. This can be expanded on if required.

Reviewer 3

1. **Response:** Summary: This paper aims for automatic prostate localization in ultrasound using SOTA detection models such as Yolo, RetinaNet, and FasterRCNN. The evaluations on three datasets demonstrates the models' high accuracy in detection and estimation of prostate volume across multiple performance metrics. The results also demonstrate the accuracy for the real-time tracking of the prostate using SOTA detection models. In terms of weakness, the paper lacks several critical discussion and evaluations that would strengthen its novelty and clinical relevance.

Response: No changes necessary.

2. **Comment:** This paper shows limited novelty beyond technical implementation of known models - Object detection is a well-investigated area in medical imaging even for prostate localization. Adapting existing technical models (Yolo, FasterRCNN etc.) to ultrasound imagery may lack sufficient novelty. It also lacks a thorough discussion of the clinical workflow integration and decision support implications.

Response: The majority of prostate localisation is either for TRUS or MR images, not SUS images. A section could be added to explain how the workflow could be used to aid clinicians in finding the prostate (since real-time results are possible) and estimating its size.

3. **Comment:** The paper lacks discussion and justification with existing research related - There are already studies leveraging object detection for prostate localization and volume estimation, including Yolo-based methods. For example, related work like [1] could be discussed and critically compared.

Response: While YOLO was used in the suggested reference, they were detecting prostate cancer in (RGB?) tissue images. The current manuscript does not detect prostate cancer, it detects the prostate as part of a workflow that will be used in risk-stratification/screening protocols, prior to detecting prostate cancer.

4. **Comment:** The authors claim the potential of using technical models to enhance clinical workflows. It is also suggested to include clinical evaluation or user study to support this claim. Showing how the tool affects e.g. clinical decision-making and time for scanning would strengthen its practical relevance.

Response: This would require quite a big change to the study. Not sure if that can be done at present.

5. **Re Comment sponse:** For experiments, the paper reports across several performance metrics (IoU, percision, Pearson correlation coefficient, etc.). It lacks statistical analysis to determine whether differences across models or datasets are significant. E.g., Bland–Altman (BA) plots can be used and analysed to assess agreement of models beyond correlation.

Response: This can be added (BA and statistical analysis).